# SliceIt!: Simulation-Based Reinforcement Learning for Compliant Robotic Food Slicing

Cristian C. Beltran-Hernandez[1,*], Nicolas Erbetti[1,*], Masashi Hamaya[1]

*Abstract*— Cooking robots have the potential to greatly enhance the home experience by automating food preparation tasks. However, enabling a robot to safely and dexterously manipulate kitchen tools like knives while handling delicate food items poses significant challenges. This study tackles the problem of training a robot arm to perform robust and compliant slicing motions on food items with varying material properties. We present SliceIt!, a simulation-based framework for training robust food-slicing skills through reinforcement learning before deployment on the physical robot. Our approach follows a real-to-sim-to-real pipeline: first collecting a small dataset of real food-cutting examples, then calibrating high-fidelity simulations of knife-food cutting interactions and robot motion control. Reinforcement learning agents are trained in this calibrated simulation environment to learn optimal compliance control policies that modulate knife forces. The learned policies are then transferred to the real robot, enabling it to perform intricate food-slicing tasks efficiently and safely by leveraging simulation-based policy training while minimizing real-world training risks, effort, and food waste.

## I. INTRODUCTION

Cooking robots, which can safely work alongside humans, hold the potential to enhance home environments and ease daily chores. Tasks such as food slicing require the robot to skillfully and safely manipulate a knife. Our research focuses on enabling a robot to learn food-cutting tasks.

Cutting skills, such as chopping and slicing, require manipulating the knife and carefully responding to the reaction forces that are exerted by the material and by the cutting board [1]. In particular, it is important for the robot to be able to adapt to the widely varying physical properties of each food product [2]. Learning-based methods are promising approaches to autonomously learning complex robotic behaviors, such as the one required for food slicing. However, such methods often need many interactions to learn useful control policies, which could result in a lot of food waste for food-slicing tasks. Furthermore, it could be dangerous to train the robot directly in the real world when the robot is handling hazardous tools such as a kitchen knife, as part of its learning paradigm involves the random exploration of actions. Thus, learning in a simulation environment is a viable solution where exploratory actions can be conducted safely.

Learning in simulation has its own challenges, namely the reality gap [3]. The difference between the simulation environment and the real world can render the learned

* Equal contribution.

This work was supported by KAKENHI Grant Number 21H04910.

[1]OMRON SINIC X Corporation, Tokyo, Japan. Corresponding author
`cristian.beltran@sinicx.com`

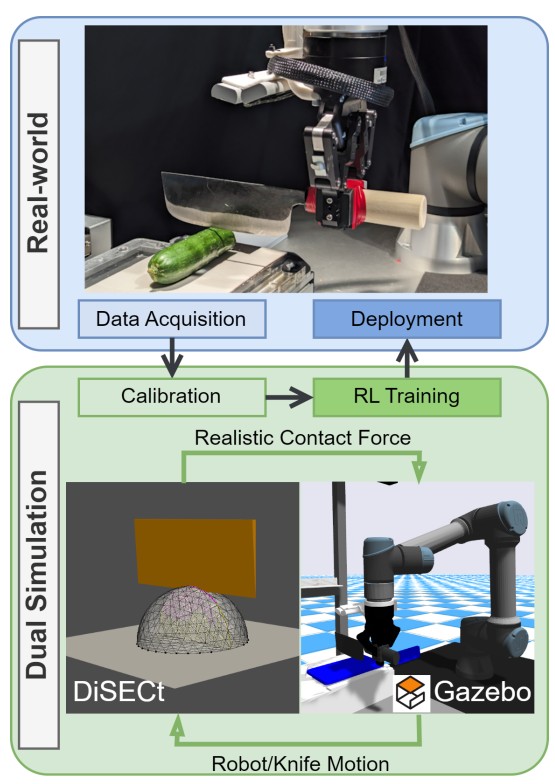

Fig. 1. Overview of proposed learning-based robot cutting framework, comprising four key stages: 1) Data collection on the real robot. 2) Calibration of the cutting simulator, DiSECt. 3) Learning a control policy within a dual simulation environment using Gazebo and DiSECt. 4) Deployment to the real robot.

control policy unusable in the real world. Recently, advanced simulators like DiSECt [4] have been introduced, offering a highly realistic representation of soft material cutting. These simulators can potentially bridge the reality gap. However, to facilitate more accessible evaluation in real-world scenarios, creating an interface between the specialized cutting simulation and real robots is essential. The existing simulators focus on the physical interactions between the knife and the object but do not consider whether the robot can feasibly realize the knife motion.

To address this challenge, we propose a dual simulation environment combining the cutting simulator (CutSim) with a robotics simulator (RoboSim). The CutSim provides more realistic interaction forces to the RoboSim. Meanwhile, the RoboSim provides the knife motion generated by the simulated robot to the CutSim. Additionally, the RoboSim is more practical to use since the motions generated can be more eas-

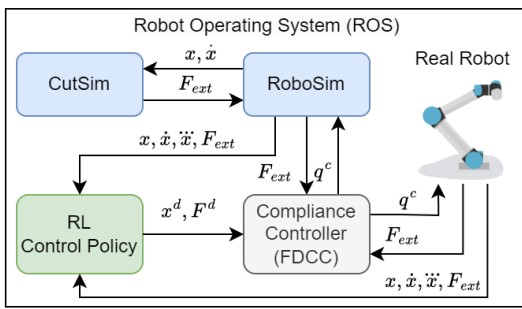

Fig. 2. ROS-powered proposed system for learning robotic cutting tasks using a dual simulation environment, reinforcement learning, and compliance control.

ily integrated into the real robot [5]. In this work, we present SliceIt!, a framework for safely learning robot cutting, that features a dual simulation environment. Our system follows a real-to-simulation-to-real (real2sim2real) [6] formulation and consists of (1) data collection from slicing real food items. (2) Calibration of the CutSim to simulate with high fidelity the collected slicing data. (3) Learning a control policy using our calibrated dual simulation environment. (4) Deployment of the policy on the real robotic system. An overview of our proposed method is depicted in Figure 1.

## II. METHODOLOGY

### A. System Overview

We introduce a robot learning system designed for food-slicing tasks, employing a real2sim2real [6] approach. In Figure 1, we outline the core elements of our proposed method.

First, the "real2sim" phase involves data acquisition and calibration of our dual simulation environment. Our simulation environment consists of two concurrent simulators: a physics simulator tailored for cutting soft materials (CutSim), and a robotic simulator (RoboSim). We gather data by having the real-world robot slice food items. This data is then utilized to fine-tune the simulation parameters of the cutting simulator. Only a few data samples are required for the calibration.

Second, the "sim2real" phase focuses on training a control policy using Reinforcement Learning (RL) within the simulation and deploying it in the real world. The combined cutting simulator and the robotic simulator are used for this purpose.

The following sections describe in detail the components of our method, the dual simulation environment, the calibration of our simulation environment, and the learning-based compliance control policy.

### B. Dual Simulation Environment

*1) CutSim:* In this work, the DiSECt simulator was used. DiSECt is a differentiable physics simulator for cutting soft materials [4]. The simulator augments the finite element method (FEM) with a continuous contact model based on signed distance fields (SDF), as well as a continuous damage model that inserts springs on opposite sides of the cutting plane and allows them to weaken until zero stiffness to model crack formation. DiSECt was chosen because it allows us to realistically simulate food cutting by calibrating its simulation parameters. The differentiability of the simulator enables the calibration of the simulation parameters using gradient-based optimization methods [4].

*Calibration*: The calibration process involves simulating the robot's cutting actions, including motion and contact force, and adjusting the simulation parameters until the force profile matches the desired one. The simulator's differentiability allows us to fine-tune these parameters using gradient-based optimization methods [4]. However, gradient-based optimization can be computationally intensive, and inappropriate initial parameters sometimes cause learning instability. Therefore, in this study, we propose a two-step approach. Initially, a non-gradient-based optimization method is used to quickly and cost-effectively identify an initial set of simulation parameters. Subsequently, a gradient-based optimization method, specifically the Adam algorithm [7], is employed to further refine these simulation parameters. To optimize the initial simulation parameters, we utilize the Tree-structured Parzen Estimator algorithm [8] as implemented in Optuna [9].

*2) RoboSim:* We use the Gazebo simulator, which is an open-source 3D robotics simulator [5]. Our choice of the Gazebo simulator was motivated by its compatibility with the Robot Operating System (ROS) [10]. ROS is an open-source robotics middleware that facilitates the integration of different robotic components. In this work, ROS was used to integrate both simulators, the real robots, our proposed RL control policy, and its low-level compliance controller as depicted in Figure 2.

### C. Learning slicing with reinforcement learning

Traditionally, fine-tuning a compliance controller for a given task is a time-consuming process that requires human expertise. To reduce these requirements, we use RL to learn the motion of the cutting action as well as the control parameters of the FDCC, inspired by previous work [11]. Compared to [11], we use FDCC because it provides better stability in singularities and requires fewer parameters to learn. As described in Figure 2, The actions of the RL agent provide the reference trajectory $x^c$ and the control parameters $[K^c, K^p, K^d]$ to the FDCC at a low control frequency. Then the FDCC directly controls the robot with joint commands at the highest control frequency available. The feedback from the robot is the knife pose, computed using forward kinematics from the robot's joint positions and the sensed force and momentum. The agent's observations consist of the knife's position relative to a target position, its velocity, and jerkiness, as well as the previous action taken and the history of $n$ force-torque readings from the sensor.

The SAC's actor-network architecture consists of a Temporal Convolutional Network (TCN) [12], that processes a history of $n$ force/torque readings from the sensor, and a fully connected network that processes the remaining

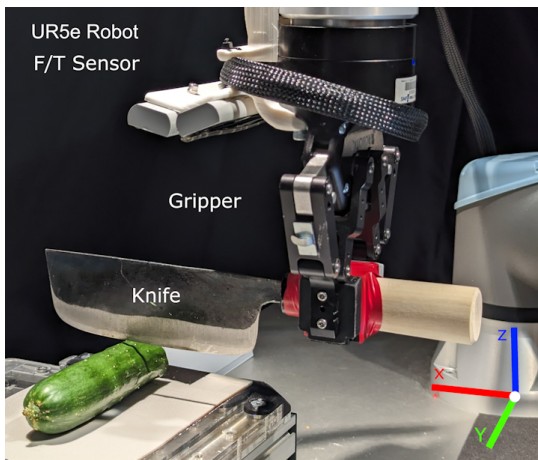

Fig. 3. Experimental setup with the real robot.

TABLE I
REAL-WORLD EXPERIMENTAL CONDITIONS

| Food Item | # of Slices | Slice size (mm) |
|-----------|-------------|-----------------|
| Cucumber | 15 | 5 |
| Tomato | 5 | 5 |
| Potato | 10 | 3 |
| Carrot | 5 | 5 |

observations. Both networks output 64 features that are concatenated and processed together on an additional fully connected network. The choice of network architecture is based on results from previous work [13].

## III. EXPERIMENTS AND RESULTS

The following experiments were designed to answer these questions:

- How well does our system perform in the real world when it has only been trained in a simulated environment?
- When it comes to the robotic cutting task, does using a more realistic simulation environment lead to better performance compared to using a less detailed one?

For the latter, we evaluate the performance of our proposed system against a baseline. This baseline involves calibrating the simulation environment and training a policy exclusively using the Gazebo Simulator [5].

### A. Experimental Setup

*1) Robotic Platform:* Our robotic platform is based on the dual-arm system initially introduced in [14]. It comprises two Universal Robot UR5e robotic arms, each equipped with a parallel gripper and a force-torque sensor positioned at the arm's end. In the context of this research, one of these robots serves a supporting role, such as holding the vegetable, while the second robot is responsible for executing the cutting task. The experimental setup is depicted in Figure 3. Notably, the robot engaged in the task is equipped with a Robotiq 85 parallel gripper featuring a custom finger adapter that enables the attachment and detachment of a kitchen knife.

*2) Calibration:* In this study, we gathered data from three distinct food items, each characterized by different material properties, to ensure diversity within our training dataset. Specifically, we selected a tomato, a cucumber, and a potato, representing items with low, medium, and high stiffness, respectively. The collected data consists of the knife's motion and the contact force applied during the cutting action. For simplicity, we maintained a constant knife speed throughout the experiments.

In the baseline case, denoted as *Gazebo only*, the cutting action was simulated using a compression spring. In the simulator, the compression spring is defined as a prismatic joint where the force constant is determined by specific simulation parameters, namely, the Error Reduction Parameter (ERP) and Constraint Force Mixing (CFM). These parameters were calibrated so that the force required to compress the spring to its maximum matches the maximum force observed in the real-world force profile, as depicted in Figure 4.

*3) Cutting Task:* In this study, the robot cutting task is defined as a single slice of the food item, which can then be executed multiple times as necessary. The goal of the robot is to maximize speed and minimize contact force and motion jerkiness. In particular, the task includes minimizing the force exerted on the cutting board with the knife. Four food items were used for validation: a cucumber, a tomato, a potato, and a carrot. As mentioned above, the first three were used for calibrating the simulation environments, while the carrot was used to evaluate the generalization capabilities of our method.

After calibration of both our method and the baseline using all of the food items, an RL agent was trained in each simulation environment for 80K time steps. The training included domain randomization by loading one of the calibrated food items into the CutSim as well as injecting a small uniformly sampled noise into the simulation parameters. A similar noise was injected into the baseline. The learning converges at around 60K time steps.

### B. Real World Experiments

The evaluation of the learned control policies involved slicing each food item multiple times. To minimize food waste, only one unit of each food item was utilized in these experiments. The specific experimental conditions are presented in Table I.

Figure 5 and 6 illustrate the obtained results. In Figure 5, we compare the contact force observed during the slicing actions for each food item between our method and the baseline. These results include the contact force exerted not only on the food item but also on the cutting board. Notably, our method consistently outperforms the baseline by applying significantly lower force during the execution of slicing actions across all tasks. Remarkably, even in the case of the carrot, which was not part of the training dataset, our proposed method demonstrated superior performance when compared to the baseline.

Figure 6 centers on the experiments related to cucumber slicing. In this visualization, the grey region corresponds to the slicing of the vegetable, while the yellow region

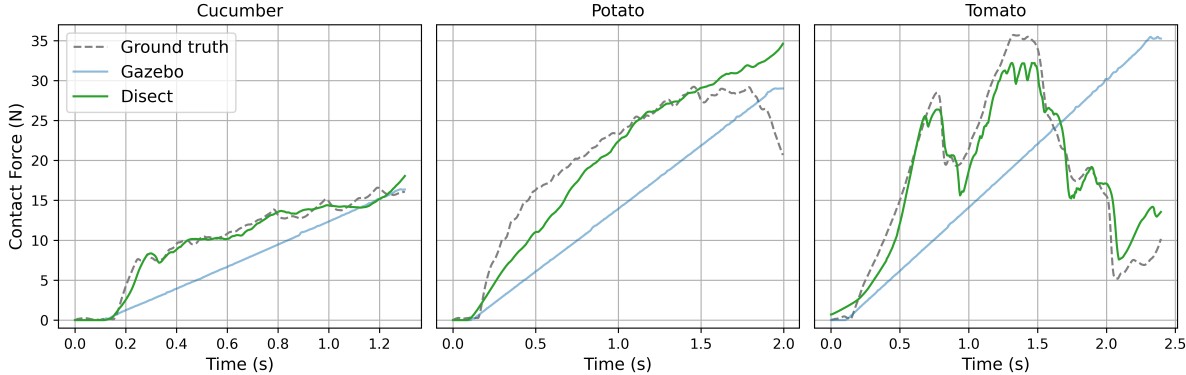

Fig. 4. Force profile of the simulated cutting after the calibration process for both simulators.

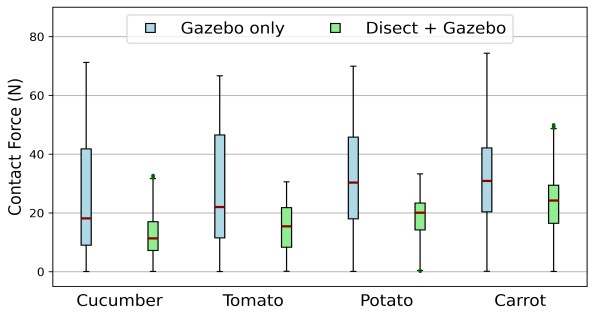

Fig. 5. Evaluation on the real robot: Slicing contact force for four vegetables, cucumber, tomato, potato, and carrot.

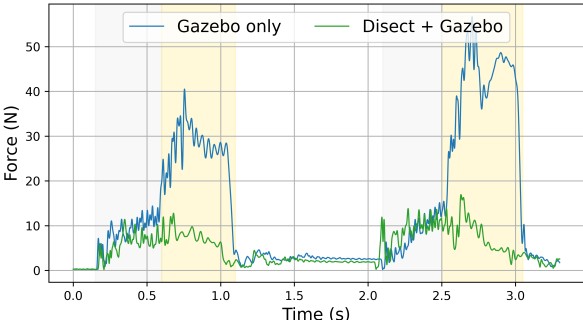

Fig. 6. Evaluation on the real robot: Normal contact force at the knife while slicing the cucumber twice. The gray region corresponds to slicing the vegetable, while the yellow region shows the contact with the cutting board.

represents the contact force applied to the cutting board. These findings suggest that the policy acquired through our proposed method achieves superior performance by exhibiting a more adept response to the abrupt stiffness transition between the vegetable and the cutting board. Similar results are observed across all tasks. These results indicate that the policy learned with our proposed method performed better by more skillfully reacting to the sudden change of stiffness, between the vegetable and the cutting board. Figure 6 centers on the experiments related to cucumber slicing. In this visualization, the grey region corresponds to the slicing of the vegetable, while the yellow region represents the contact force applied to the cutting board. The results show clearly that our proposed method performs better than the baseline by more skillfully reacting to the change of stiffness between the food item and the cutting board. These results are consistent across all trials and tasks.

## IV. CONCLUSION

In this study, we introduced SliceIt! a learning-based robotic system for handling food-cutting tasks. Our system combines two key components: a dual simulation environment, and a control policy based on Reinforcement Learning. The aim is to enable a collaborative robot (cobot) or industrial robot arms to perform these tasks safely and accurately by adapting to varying conditions using compliance control.

One of the advantages of using our proposed method is the reduction of food waste while learning the control policies, as only a few real-world samples are required. The experimental results support our hypothesis that using a highly realistic simulation environment is beneficial to learning safer control policies.

One limitation of our approach is the increased computation time when using a highly realistic cutting simulator compared to a simplified one. In our experiments, training the RL policy using our method took approximately 40 hours in total, whereas using Gazebo alone required only about 4 hours. This discrepancy arises from the more demanding computational requirements for each simulation time step in DiSECt. However, the additional computation time proves to be worthwhile, given the significant improvement in real robot performance achieved. A promising area for future research involves finding ways to reduce the extensive computational time.

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
