# OpenReview forum: "SliceIt!: Simulation-Based Reinforcement Learning for Compliant Robotic Food Slicing"
_IEEE.org/2024/ICRA/Workshop/CookingRobot — CookingRobot2024 Poster_

### Official Review · Reviewer_ttfQ · 2024-04-11
**Review of "SliceIt!: Simulation-Based Reinforcement Learning for Compliant Robotic Food Slicing"**

**Rating:** 8
**Confidence:** 4

**Review:**

This paper proposes the whole framework of simulation-based RL for robotic food slicing, including system identification of DiSECt, RL in DiSECt, and deployment on the actual robot.

Major Comemnts
* This paper shows the novel framework for simulation-based slicing learning.
* The concept is intriguing; however, there is room for consideration regarding how essential such a system truly is for cooking.

Video
* This is a very good video. I'm also interested in the results with more vegetables, as well as with meat and fish.

---

### Official Review · Reviewer_bzeJ · 2024-04-15
**Review of "SliceIt!: Simulation-Based Reinforcement Learning for Compliant Robotic Food Slicing"**

**Rating:** 9
**Confidence:** 4

**Review:**

The paper proposes a dual simulation reinforcement learning approach, combining a cutting simulator for realistic interaction forces and a robotic simulator for easier integration onto a physical robot for a food slicing task. It validates the proposed method compared to baseline results on the physical robot.

Comments on Paper:
* The paper addresses an important problem in cooking where learning on real objects is infeasible due to concerns regarding food waste or safety, while handling soft materials is often challenging in robotic simulation.
* The method of using a high-fidelity cutting simulator for accurate force interaction in soft object manipulation while using a robot simulator for easier integration onto a physical robot is novel and sound.
* The experiment of food slicing and comparison against the baseline is well executed with physical robots.
* The authors could discuss how the proposed method scales when using more variations of knives and food items or when the cutting motions involve more complexity (e.g., not just pressing down but including slicing motions going back and forth). Does it require more real data for calibration? Does it increase computation time? How does it affect convergence?

Comments on Video:
* The video provides a nice summary of the method, experiment, and results.
* It would be interesting to see a failure case or how the baseline performs the task. In Fig. 6, there is a large contact force when in contact with the cutting board when using only Gazebo. However, it is non-intuitive how the improvement of 3-5N during cutting affects the cutting performance.